# A theoretical model of factors influencing online consumer purchasing behavior through electronic word of mouth data mining and analysis

**Qiwei Wang[1], Xiaoya Zhu[2]\*, Manman Wang[1], Fuli Zhou[1], Shuang Cheng[1]**

**1** School of Economics and Management, Zhengzhou University of Light Industry, High-tech District, Zhengzhou City, Henan Province, China, **2** School of Politics and Public Administration, Soochow University, Gusu District, Suzhou City, Jiangsu Province, China

\* zhuxiaoya@suda.edu.cn

**Data Availability Statement:** All relevant data are within the manuscript and its Supporting Information files.

## Abstract

The coronavirus disease 2019 pandemic has impacted and changed consumer behavior because of a prolonged quarantine and lockdown. This study proposed a theoretical framework to explore and define the influencing factors of online consumer purchasing behavior (OCPB) based on electronic word-of-mouth (e-WOM) data mining and analysis. Data pertaining to e-WOM were crawled from smartphone product reviews from the two most popular online shopping platforms in China, Jingdong.com and Taobao.com. Data processing aimed to filter noise and translate unstructured data from complex text reviews into structured data. The machine learning based K-means clustering method was utilized to cluster the influencing factors of OCPB. Comparing the clustering results and Kotler's five products level, the influencing factors of OCPB were clustered around four categories: perceived emergency context, product, innovation, and function attributes. This study contributes to OCPB research by data mining and analysis that can adequately identify the influencing factors based on e-WOM. The definition and explanation of these categories may have important implications for both OCPB and e-commerce.

## 1. Introduction

A prolonged quarantine and lockdown imposed by the coronavirus disease 2019 (COVID-19) pandemic has changed the human lifestyle worldwide. The COVID-19 pandemic has negatively impacted various sectors such as manufacturing, import and export trade, tourism, catering, transportation, entertainment, especially retail and hence the global economy. Consumer behavior has gradually shifted toward contactless services and e-commerce activities owing to the COVID-19 [1].

Consumers are relying on e-commerce more than ever to protect their health. Recent advances in information technology, digital transformation, and the Internet helped consumers to encounter the COVID-19 to meet the needs of the daily lives, which led to an increase in the importance of e-commerce and changes in consumers' online purchasing patterns [2].

**Funding:** The author(s) disclosed receipt of the following financial support for the research, authorship, and/or publication of this article: This study was supported by the Henan Province Philosophy and Social Science Planning Project (grant number. 2020CZH012), the Henan Key Research and Development and Promotion Special (Soft Science Research) (grant number. 222400410126), the Jiangsu Province Social Science Foundation Youth Project (grant number. 21GLC012) and the Doctor Fund of Zhengzhou University of Light Industry (grant number. 2020BSJJ022, 2019BSJJ017). The funders had no role in study design, data collection and analysis, decision to publish, or preparation of the manuscript.

**Competing interests:** The authors have declared that no competing interests exist.

When consumers shop online, their behavior is considered non-traditional, and is illustrated by a new trend and current environment. To analyze the influencing factors of online consumer purchasing behavior (OCPB), it is necessary to consider several factors, such as the price and quality of a product, consumers' preferences, website design, function, security, search, and electronic word-of-mouth (e-WOM) [3]. As the current website design and payment security have become a user-friendly and guaranteed system compared with a decade ago, some factors are no longer considered as essential. By contrast, greater diversity and complexity have become the main characteristics of the influencing factors. Furthermore, under the traditional sales model, consumers' purchase decisions were simple, while online consumers have more options in terms of shopping channels and decision choices. Meanwhile, in recent years, consumers' preferences have gradually shifted from standardized products to customized and personalized. In line with these changes, information technology and data science, such as big data analytics, data mining from e-WOM, and machine learning (ML), adaptively analyze data regarding online consumers' needs to obtain more accurate data.

Since the concept of big data was proposed in 2008, it has been applied and developed lasting 14 years, emerging as a valuable tool for global e-commerce recently. However, most enterprises have failed to seize the benefits generated from big data. In the context of big data, a huge number of comments were posted regarding e-malls (Amazon, Taobao, etc.) and online social media (blogs, Bulletin Board System, etc.). For instance, Amazon was the first e-commerce company to establish an e-WOM system in 1995, which provided the company with valuable suggestions from online consumers. E-WOM has greater credibility and persuasiveness, compared with traditional word of mouth (WOM), which is limited by various subjective factors. Moreover, e-WOM has the advantage of containing not only structured data (e.g., ratings) but also unstructured data (e.g., the specific content of consumer reviews). However, e-WOM provides product-related information that cannot be directly transformed to a research objective. Thus, an innovative method of big data analytics needs to be utilized to explore the influencing factors of OCPB, which shows the advantage of interdisciplinary applications.

The research problems are to explore the factors influencing OCPB through e-WOM data mining and analysis and explain the most important influencing factors for online consumers that are likely to exist in the future within the context of the COVID-19. The study fulfills the literature gaps on exploring influencing factors of OCPB from the perspective of e-WOM. The study makes a significant contribution to the consumer study because its findings can adequately identify the influencing factors of OCPB. It also provides the theoretical and managerial implications of its findings including how e-commerce platforms can use such data to adapt their platforms and marketing strategies to diverse situations.

The remainder of this is organized as follows. Section 1 presents the introduction. Section 2 discusses the literature review and hypotheses. Section 3 provides the methodology, including data mining and analysis. Section 4 describes the results, including K-means results, performance metrics, hypotheses results, and a theoretical model. Sections 5 and 6 provide discussion and conclusion, respectively.

## 2. Literature review and hypotheses

### 2.1 Influencing factors of OCPB

Online shopping has an increasing sales volume each year, which has become huge challenges for offline retailers. Venkatesh et al. [4] found that culture, demographics, economics, technology, and personal psychology were the main antecedents of online shopping, and the main drivers of online shopping were congruence, impulse buying behavior, value consciousness, risk, local shopping, shopping enjoyment, and browsing enjoyment by a comprehensive

model of consumers online purchasing behavior. Within the context of COVID-19, OCPB is positively impacted by attitude toward online shopping [5]. Melović et al. [6] focused on millennials' online shopping behavior and noted that the demographic characteristics, the affirmative characteristics, risks and barriers of online shopping were the key influencing factors. Based on the stimulus-organism-response (SOR) theory model, consumers' actual impulsive shopping behavior is impacted by arousal and pleasure [7]. Furthermore, the influencing factors of consumers' purchase behavior toward green brands are green perceived quality, green perceived value, green perceived risk, information costs saved, and purchase intentions by perceived risk theory [8]. The positive and negative effects of corporate social responsibility practices on consumers' pro-social behavior are moderated by consumer-brand social distance, although it also impacts consumer behavior beyond the consumer-brand dyadic relationship [9]. Green perceived value, functional value, conditional value, social value, and emotional value may impact green energy consumers' purchase behavior [10]. Recipients' behavior and WOM predict distant consumers' behavior [11]. Moreover, consumer behavior is significantly impacted by financial rewards, perceived intrusiveness, attitudes toward e-mail advertising, and intentions toward the senders [12]. Store brand consumer purchase behavior is positively impacted by store image perceptions, store brand price-image, value consciousness, and store brand attitude [13]. A meta-analysis summarizes the influencing factors of consumer behavior, household size, store brands, store loyalty, innovativeness, familiarity with store brands, brand loyalty to national brands, price consciousness, value consciousness, perceived quality of store brands, perceived value for money of store brands, and search versus experience positively impact consumer behavior, whereas price–quality consciousness, quality consciousness, price of store brands, and the consequences of making a mistake in a purchase negatively impact consumer behavior [14].

Based on protection motivation theory and theory of planned behavior (TPB), consumers are more likely to use online shopping channels than offline channels during the COVID-19 pandemic [15]. The TPB is also adapted to explain the influencing factors of consumers' behavior in different areas. For instance, the attitude, perceived behavioral control, policy information campaigns, and past-purchase experiences significantly impact consumers' purchase intention, whereas subjective and moral norms show no significant relationship based on the extended TPB [16]. Although green purchase behavior has different antecedents, only personal norms and value for money have fully significant relationships with green purchase behavior, environmental concern, materialism, creativity, and green practices. Functional value positively influences purchase satisfaction, physical unavailability, materialism, creativity, and green practices, and negatively influences the frequency of green product purchase by extending the TPB [17]. Meanwhile, Nimri et al. [18] utilized the TPB in green hotels and showed that knowledge and attitudes, as well as subjective injunctive norms, positively impacted consumers' purchase intention. Yi [19] observed that attitude, social norm, and perceived behavioral control positively impacted consumers' purchase intention based on the TPB. The factors of supportive behaviors for environmental organizations, subjective norms, consumer attitude toward sustainable purchasing, perceived marketplace influence, consumers' knowledge regarding sustainability-related issues, and environmental concern are the influencing factors of consumers sustainable purchase behavior [20]. Consumers' green purchase behavior is impacted by the intention through support of the TPB [21].

## 2.2 Influencing factors of emergency context attribute

Consumers exhibited panic purchase behavior during the COVID-19, which might have been caused by psychological factors such as uncertainty, perceptions of severity, perceptions of

scarcity, and anxiety [22]. In the reacting phase, consumers responded to the perceived unexpected threat of the COVID-19 and intended to regain control of lost freedoms; in the coping phase, they addressed this issue by adopting new behaviors and exerting control in other areas, and in the adapting phase, they became less reactive and accommodated their consumption habits to the new normal [23]. The positive and negative e-WOMs may have significant influence on online consumers' psychology. Specifically, e-WOM that conveys positive emotions (pride, surprise) tends to have a greater impact on male readers' perception of the reviewer's cognitive effort than female readers, whereas e-WOM that conveys negative emotions (anger, fear) has a greater impact on cognitive effort of female readers than male readers [24]. When online consumers believe their behavioral effect is feasible and positive, while their behavioral decision is related to the behavioral outcome [25]. Traditionally, there are five stages of consumer behavior that include demand identification, information search, evaluation of selection, purchase, and post-purchase evaluation. In addition, online purchase behavior involved in the various stages can be categorized into: attitude formation, intention, adoption, and continuation. Most of the important factors that influence online purchasing behavior are attitude, motivation, trust, risk, demographics, website, etc. "Internet Adoption" is widely used as a basic framework for studying "online buying adoption". Psychological and economic structures associated with the IT adoption model can be used as the online consumer's behavior models for innovative marketers. The adoption of online purchasing behavior is explained by different classic models of attitude behavior [26]. Consumer behaviors represented by customer trust and customer satisfaction, influence repurchase and positive WOM intentions [27]. Return policy leniency, cash on delivery, and social commerce constructs were significant facilitators of customer trust [28]. Meanwhile, seller uncertainty was negatively influenced by return policy leniency, information quality, number of positive comments, seller reputation, and seller popularity [29]. Social commerce components were a necessity in complementing the quality dimensions of e-service in the environment of e-commerce [30]. Perceived security, perceived privacy and perceived information quality were all significant facilitators of online customer trust and satisfaction [31].

E-service quality, consumer social responsibility, green trust and green perceived value have a significant positive impact on green purchase intention, whereas greenwashing has a significant negative impact on green purchase intention. In addition, consumer social responsibility, green WOM, green trust and green perceived value positively moderated the relationship between e-service quality and green purchase intention, while greenwashing and green participation negatively moderated the relationships [32]. Large-scale online promotions provide mobile users with a new shopping environment in which contextual variables simultaneously influence consumer behavior. There is ample evidence suggesting that mobile phone users are more impulsive during large-scale online promotion campaigns, which are the important contextual drivers that lead to the occurrence of mobile users' impulse buying behavior in the "Double 11" promotion. The results show that promotion, impulse buying tendency, social environment, aesthetics, and interactivity of mobile platforms, and available time are the key influencing factors of impulse buying by mobile users [33]. Environmental responsibility, spirituality, and perceived consumer effectiveness are the key psychological influencing factors of consumers' sustainable purchase decisions, whereas commercial campaigns encourage young consumers to make sustainable purchases [34]. The main psychological factors affecting consumers' green housing purchase intention include the attitude, perceived moral obligation, perceived environmental concern, perceived value, perceived self-identity, and financial risk. Subjective norms, perceived behavioral control, performance risk, and psychological risk are not included. Meanwhile, the purchase intention is an important predictor of consumers' willingness to buy [35]. The perceived control of flow and focus will positively

affect the utilitarian value of consumers, while focus and cognitive enjoyment will positively impact the hedonic value. Moreover, utilitarian value has a greater impact on satisfaction than hedonic value. Finally, hedonic value positively impacts unplanned purchasing behavior [36]. Utilitarian and hedonic features achieve high purchase and WOM intentions through social media platforms and also depend on gender and consumption history [37].

Therefore, we present the following hypothesis:

Hypothesis 1 (H1): Perceived emergency context attribute is the influencing factor of OCPB.

## 2.3 Influencing factors of perceived product attribute

Product quality and preferential prices are the major factors considered by online consumers, especially within the context of the COVID-19. Specifically, online shopping offers lower price, more choices for better quality products, and comparison between them [1]. Under the circumstance of online reviews, an original equipment manufacturer (OEM) selling a new product carefully decides whether to adopt the first phase remanufacturing entry strategy or to adopt the phase 2 remanufacturing entry strategy under certain conditions. Meanwhile, the OEM adopts penetration pricing for new and remanufactured products, when the actual quality of the product is high. Otherwise, it adopts a skimming pricing strategy, which is different from uniform pricing when there are no online reviews. Online reviews significantly impact OEM's product profits and consumer surplus. Especially when the actual quality of the product is high enough, the OEM and the consumer will be also reciprocal [38]. Online reviews reduce consumers' product uncertainty and improve the effect of consumer purchase decisions [39, 40]. Uzir et al. [41] utilized the expectancy disconfirmation theory to prove that product quality positively impacts customer satisfaction, while product quality and customer satisfaction are mediated by customer's perceived value. Product quality and customer's perceived value will have greater influence with higher frequency of social media use. Nguyen et al. [42] studies consumer behavior from a cognitive perspective, and theoretically develops and tests two key moderators that influence the relationship between green consumption intention and behavior, namely the availability of green products and perceived consumer effectiveness.

Both sustainability-related and product-related texts positively influence consumer behavior on social media [43]. Online environment, price, and quality of the products are significantly impacted by OCPB. Godey et al. [44] explained the connections between social media marketing efforts and brand preference, price premium, and loyalty. Brand love positively impacts brand loyalty, and both positively impact WOM and purchase intention [45]. Brand names have a systematic influence on consumer's product choice, which is moderated by consumer's cognitive needs, availability of product attribute information, and classification of brand names. In the same choice set, the share of product choices with a higher brand name will increase and be preferred even if it is objectively inferior to other choices. Consumers with low cognitive needs use the heuristic of "higher is better" to select options labeled with brand names and choose brands with higher numerical proportions [46].

Therefore, we present the following hypothesis:

Hypothesis 2 (H2): Perceived product attribute is the influencing factor of OCPB.

## 2.4 Influencing factors of perceived innovation attribute

Product innovation increases company's competitive advantage by attracting consumers, whereas the enhancement of innovative design according to consumer behavior accelerates

the development of sustainable product [47, 48]. The innovation, WOM intentions and product evaluation can be improved positively by emotional brand attachment and decreased by perceived risk [49]. Based on the perspective of evolutionary, certain consumer characteristics, such as buyer sophistication, creativity, global identity, and local identity, influence firms' product innovation performance, which can increase the success rate of product innovation, and enhance firms' research and development performance [50]. However, technological innovation faces greater risk as it depends on market acceptance [51]. Moreover, electronic products rely more on technological innovation compared with other products, which maintain the profit and market [52]. The technological innovation needs to apply logical plans and profitable marketing strategies to reduce consumer resistance to innovation. Thus, Sun [53] explains the relationship between consumer resistance to innovation and customer churn based on configurational perspective, whereas the results show that response and functioning effect are significant but cognitive evaluation is not.

Based on the perspective of incremental product innovation, aesthetic and functional dimensions positively impact perceived quality, purchase intention, and WOM, whereas symbolic dimension only positively impacts purchase intention and WOM. By contrast, aesthetic and functional dimensions only positively impact perceived quality, whereas symbolic dimension positively impacts purchase intention and WOM. Furthermore, perceived quality partially mediates the relationship between aesthetic and functional dimensions and purchase intention and WOM by incremental product innovation, whereas perceived quality fully mediates the relationship between aesthetic and functional dimensions and purchase intention and WOM by radical product innovation [54]. Contextual factors, such as size of organizations and engagement in research and development activity, moderate the relationship between design and product innovation outcomes [55]. For radical innovations, low level of product innovation leads to more positive reviews and less inference of learning costs. As the functional attribute of radical innovations is not consistent with existing products, it is difficult for consumers to access relevant product category patterns and thus transfer knowledge to new products. The product innovation of aesthetics, functionality, and symbolism positively impact willingness to pay, purchase intention, and WOM through brand attitude [56]. This poor knowledge transfer results in consumers feeling incapable of effectively utilizing radical innovations, resulting in greater learning costs. In this case, product designs with low design novelty can provide a frame of reference for consumers to understand radical innovations. However, incremental product innovation shows no significant difference between a low and high level of design newness [57].

Therefore, we present the following hypothesis:

Hypothesis 3 (H3): Perceived innovation attribute is the influencing factor of OCPB.

## 2.5 Influencing factors of perceived motivation attribute

The research has proven that almost all consumers' purchases are motivated by emotion. Under this circumstance, an increase in online consumers' positive emotions increases, their purchase frequency, whereas an increase in online consumers' negative emotions reduces their purchase frequency. Additionally, user interface quality, product information quality, service information quality, site awareness, safety perception, information satisfaction, relationship benefits and related benefit factors have negative impacts on consumers' online shopping emotionally. Nevertheless, only product information quality, user interface quality, and safety perception factors have positive effects on online consumer sentiment [58]. E-WOM carries emotional expressions, which can help consumers express the emotions timely. Pappas et al.

[59] divides consumers' motivation into four factors, namely entertainment, information, social-psychological, and convenience, while emotions into two factors, namely positive and negative. Specifically, according to complexity and configuration theories, a conceptual model by a fuzzy-set qualitative comparative analysis examines the relationship between a combination of motivations, emotions, and satisfaction, while results indicate that both positive and negative emotions can lead to high satisfaction when combing motivations.

From the perspective of SOR theory, consumers' motivation is greatly influenced by self-consciousness, while conscious cognition plays the role of intermediary. First, after being stimulated by the external environment, online consumers will form "cognitive structure" depending on their subjectivity. Instead of taking direct action, they deliberately and actively obtain valid information from the stimulus process, considering whether to choose the product, and then react. Second, the stimulation stage in the retail environment can often attract the attention of consumers and cause the change of their psychological feelings. This stimulation is usually through external environmental factors, including marketing strategies and other objective influences. Third, organism stage is the internal process of an individual. It is a consumers' cognitive process about themselves, their money, and risks after receiving the information they have seen or heard. Reaction includes psychological response and behavioral response, which is the decision made by the consumer after processing the information [60]. Based on literature review, 10 utilitarian motivation factors, such as desire for control, autonomy, convenience, assortment, economy, availability of information, adaptability/customization, payment services, absence of social interaction, and anonymity and 11 hedonic motivation factors, such as visual appeal, sensation seeking/entertainment, exploration/curiosity, escape, intrinsic enjoyment, relaxation, pass time, socialize, self-expression, role shopping, and enduring involvement with a product or service, are refined [61]. Consumers' incidental moods can improve online shopping decisions impulsivity, while decision making process can be divided into orientation and evaluation [62]. Sarabia-Sanchez et al. [63] combine K-means cluster and ANOVA analyses to explore the 11 motivational types of consumer values, which are achievement, tradition, inner space, universalism, hedonism, ecology, self-direction (reinforcement, creativity, harmony, and independence), and conformity.

Therefore, we present the following hypothesis:

Hypothesis 4 (H4): Perceived motivation attribute is the influencing factor of OCPB.

## 3. Materials and methods

### 3.1 Research design

Given the present study's objective to identify the influencing factors of OCPB, we analyzed e-WOM using big data analysis. To obtain accurate data of the influencing factors on OCPB, smartphones were the main object of data crawling. The rationale behind this choice is as follows. First, the time people spend using their smartphones is gradually increasing. Nowadays, smart phones are not only used for telephone calls or text messages, but also for taking photographs, recording video, surfing the web, online chatting, online shopping, and other such uses [64]. Second, smartphones have become a symbol of personal identification, as users' using fingerprint or facial scans are frequently used to unlock devices, conduct online transactions, and make reservations, etc. Finally, smartphones' software and hardware are updated frequently, so they may be considered high-tech products. Therefore, smartphones were chosen as the research object to determine which influencing factors affect OCPB.

Fig 1 shows the e-WOM data mining process and methods used. A dataset obtained from Taobao.com and Jingdong.com was collected by utilizing a Python crawling code, additional

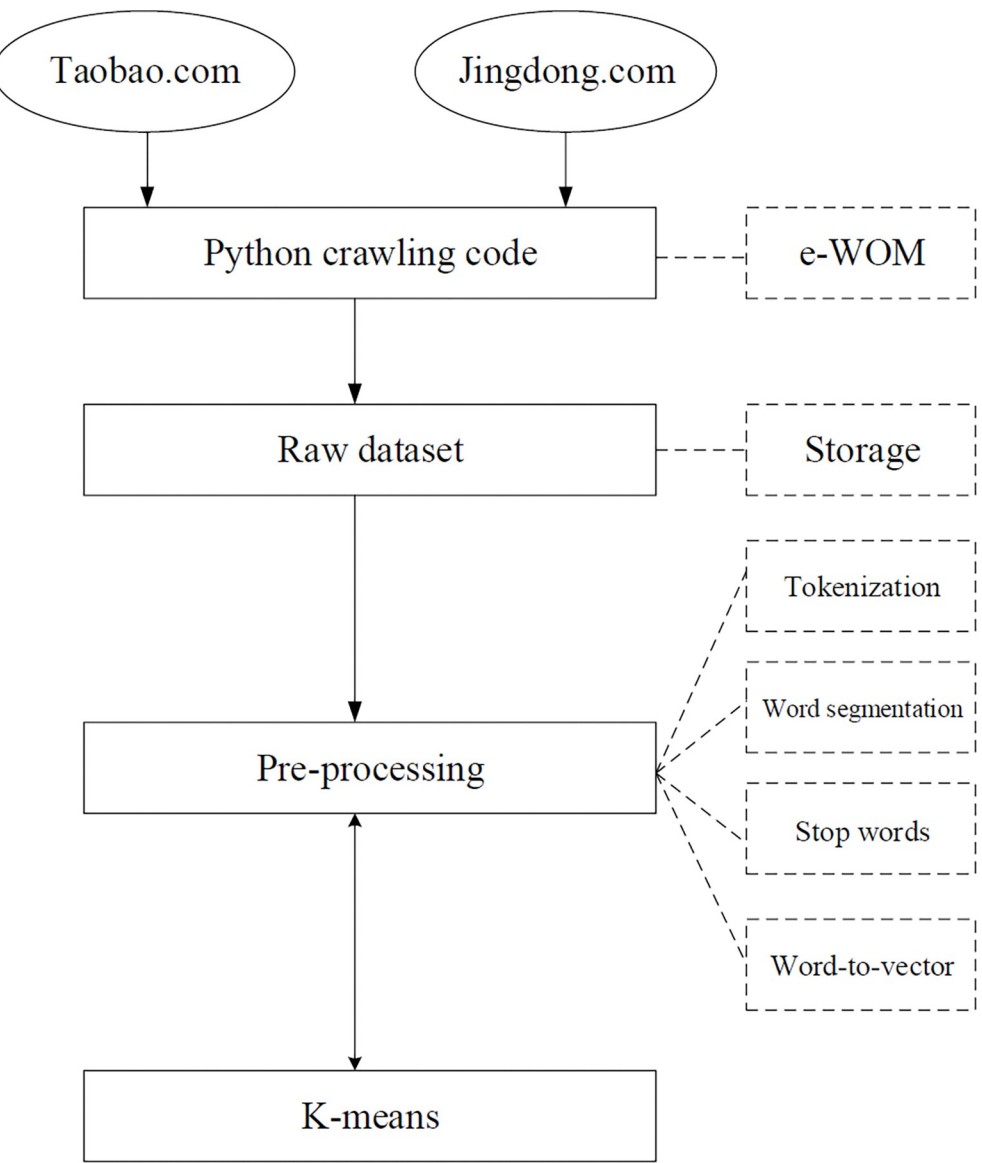

**Fig 1. Research techniques used in this study.**

details of which are provided in Section 2.3. Section 2.4 addresses issues regarding language complexity. Moreover, Section 2.5 refers to the clustering of the influencing factors of OCPB through the K-means method of ML.

## 3.2 Data collection

The data were crawled from the e-commerce platforms Jingdong.com and Taobao.com by utilizing Python software. Jingdong and Taobao are the most powerful and popular platforms in China having professional e-WOM and user-friendly review systems. Specifically, the smartphone brands selected for analysis were Apple, Samsung, and Huawei because these three smartphone companies occupy the largest percentage of the smartphone market.

The authors determined that the analysis of the influencing factors of OCPB would be more persuasive and realistic by choosing smartphone models with high usage rate and liquidity.

Thus, products reviews were crawled for the purchase of newly launched smartphones from Apple, Samsung, and Huawei in 2022. Specifically, to guarantee high-quality data, reviews from Taobao flagship stores and Jingdong directly operated stores were selected. However, we only collected reviews' text content instead of images, videos, ratings, or rankings, the rationale was to ensure the reliability of data and meet research objectives. For instance, some e-commerce sellers attempt to increase their sales volume through deceitful methods, such as by faking ratings, rankings, and positive comments. Furthermore, online sellers and e-commerce companies (rather than consumers) often decide which smartphones are highest-rated and highest-selling. Finally, nowadays, the content of online reviews is not limited to text, as they also involve pictures, videos, and ratings, which have limited contribution in analyzing influencing factors of OCPB. Thus, the analyzed data regarding e-WOM in reviews was limited to text content.

In addition, to accurately reflect the real characteristics of OCPB during the COVID-19 pandemic, the study period ranged between February and May, 2022 (4 months). During that 4-month period, consumers exhibited a preference for buying products from e-commerce platforms. Specifically, the number of text reviews for the aforementioned types of smartphones was 51,2613 and 44,3678 in Taobao and Jingdong, respectively, for a total of 956,291 reviews.

## 3.3 Textual review processing method

As the crawled data exhibited noise, several data cleaning methods were adopted to filter noise and transform unstructured data of complex contextual review into structured data. Fig 1 shows the main procedures of the reviews' pre-processing and the details are as follows.

First, to identify the range of sentences and for further data processing, sentences were apportioned using Python's tokenizer package.

Second, this study employed Python's Jieba package to perform word segmentation. The Jieba package is the Python's best Chinese word segmentation module, comprising three modes. The exact mode was used to segment the sentences as accurately as possible, so they may be suitable for textual context analysis. The full mode was used to scan and process all words in each sentence, although it had a relatively high speed, it had a low capacity to resolve ambiguity. Additionally, the search engine mode segmented long words a second time, which allowed for the improvement of the recall rate, and was suitable for engine segmentation based on Jieba's exact mode.

Third, stop words were deleted by referring to a stop words list. These included conjunctions, interjections, determiners, and meaningless words, among others. Finally, Python's Word-to-vector (Word2vec) package was imported in the next step. Word2vec is an efficient training word vector model proposed by Mikolov [65, 66]. The basic starting point was to match pairs of similar words. For instance, when "like" and "satisfy" appeared in a same context, they showed a similar vector, as both words had a similar meaning. Kim et al. [67] stated that a word could be considered a single vector and real numbers in the Word2vec model. In fact, most supervised ML models could be summarized as $f(x)->y$. Moreover, $x$ could be considered a word in a sentence, while $y$ could be considered this word in the context. Word2vec aimed to decide whether the sample of $(x,y)$ could match the laws of natural language. Namely, after the process of Word2vec, the combination of word $x$ and word $y$ could be reasonable and logical or not. Table 1 shows the results of text processing.

**Table 1. Text processing results.**

| Satisfy | Very | Screen | Earphone | Need | Sound effect |
|---|---|---|---|---|---|
| Automatic | Photograph | Super Remote distance | Powerful | Record | Moment |
| Speed | Same | Run | Frames per second | 44 | Fast |

### 3.4 Influencing factors analysis by K-means

ML styles are divided into supervised and unsupervised algorithms. This study mainly utilized unsupervised algorithms to analyze the clusters of influencing factors of OCPB. Unsupervised algorithms consist in the clustering of unknown or unmarked objects without a trained sample [68]. This study utilized K-means to cluster the influencing factors.

For a given sample set, the K-means algorithm divides the sample set into $k$ clusters according to the distance between samples. The main algorithm's logic is to make the points in the cluster as close as possible, and to make the distance between the clusters as large as possible. Assuming that clusters can be divided into $(C_1, C_2, \ldots, C_k)$, the Euclidean distance of $E$ is shown in Eq 1.

$$E = \sum_{i=1, x \in C_i}^{k} \|x - \mu_i\|_2^2 C_i, \tag{1}$$

Here, $u_i$ is the mean vector of $C_i$:

$$\mu_i = \frac{1}{|C_i|} \sum_{x \in C_i} x. \tag{2}$$

The main procedures of K-means were the following.

Step 1 consisted of inputting the samples $D = \{x_1, x_2, \ldots x_m\}$, $K$ is the number of clusters, and appears as $C = \{C_1, C_2, \ldots C_k\}$.

In Step 2, $K$ samples were randomly selected from data set $D$ as the initial $K$ centroid vectors: $\{\mu_1, \mu_2, \ldots \mu_k\}$.

In Step3, for $i = 1, 2, \ldots, m$, the distance between samples $x_i$ and centroid vectors $\mu_{j(j=1,2,\ldots,k)}$ was calculated: $d_{ij} = \|x_i - \mu_j\|_2^2$, $x_i$ is marked as the category $\lambda_j$, and $\mu_i$ is the smallest value of $d_{ij}$. Currently to update $C_{\lambda_j} = C_{\lambda_j} \cup (x_i)$.

In Step 4, for any clusters $C_j$, all samples in $(j = 1, 2, \ldots, k)$ needed to be recalculated according to the new centroid vectors: $\mu_j = \frac{1}{|C_j|} \sum_{x \in C_i} x$.

For Step5, it was necessary to repeat Steps 3 and 4, until all the centers $\mu$ remained steady. The final clustering result can be shown as $C = \{C_1, C_2, \ldots C_k\}$.

The main procedures of K-means, according to Jain [69], are shown in Table 2.

## 4. Results

### 4.1 K-means results

Based on the main procedures of K-means (Table 2), the results are presented in Figs 2–4.

**Table 2. Main procedures of K-means.**

| |
|---|
| Select $k$ users as centroids based on the dataset |
| **Input:** $u$, training users; $k$, the number of clusters |
| **Output:** $\{c_1, c_2, \ldots c_k\}$, $k$ centroids |
| 1. Determine the expected numbers of clusters, $k$ |
| 2. Select the users consistently at random from $u$, as initial starting points. |
| 3. Assign each user to the cluster with the nearest centroid. |
| 4. Calculate the mean of all clusters and update the centroid value according to the mean value of that cluster. |
| 5. Repeat Steps 3 and 4, until no user changes its cluster membership or any other convergence criteria are met. |
| 6. Return $\{c_1, c_2, \ldots c_k\}$ |

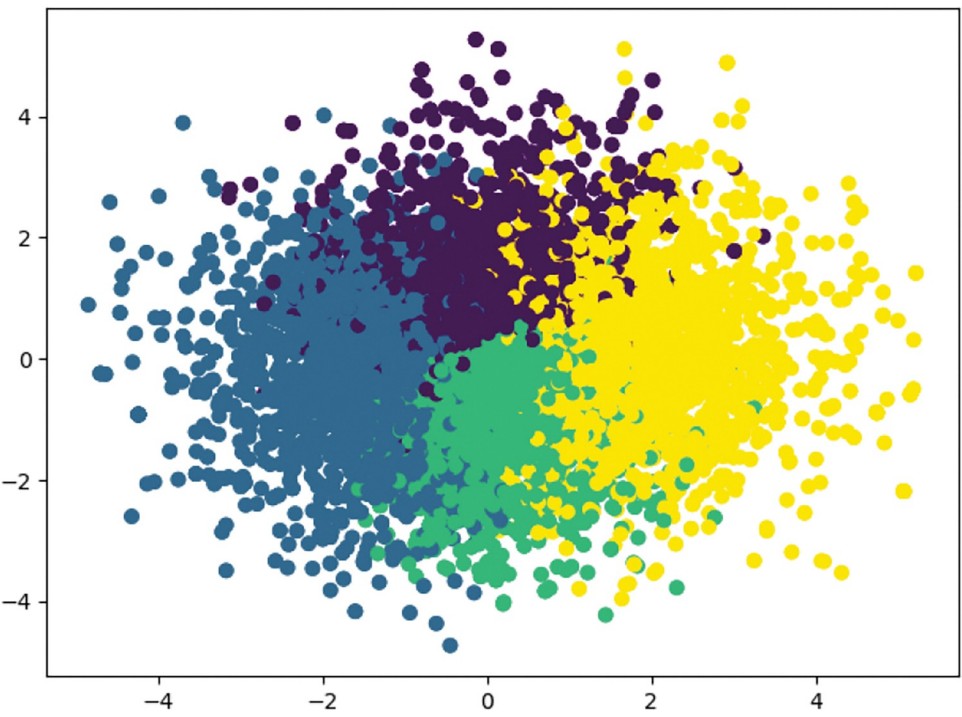

**Fig 2. Results of K-means clustering for Jingdong.**

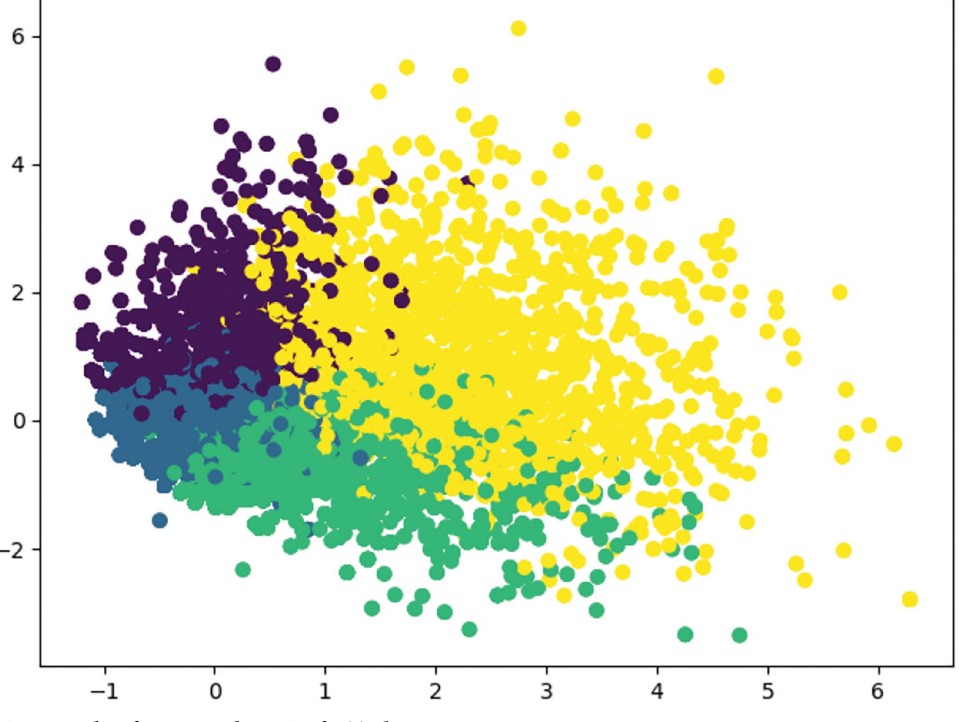

**Fig 3. Results of K-means clustering for Taobao.**

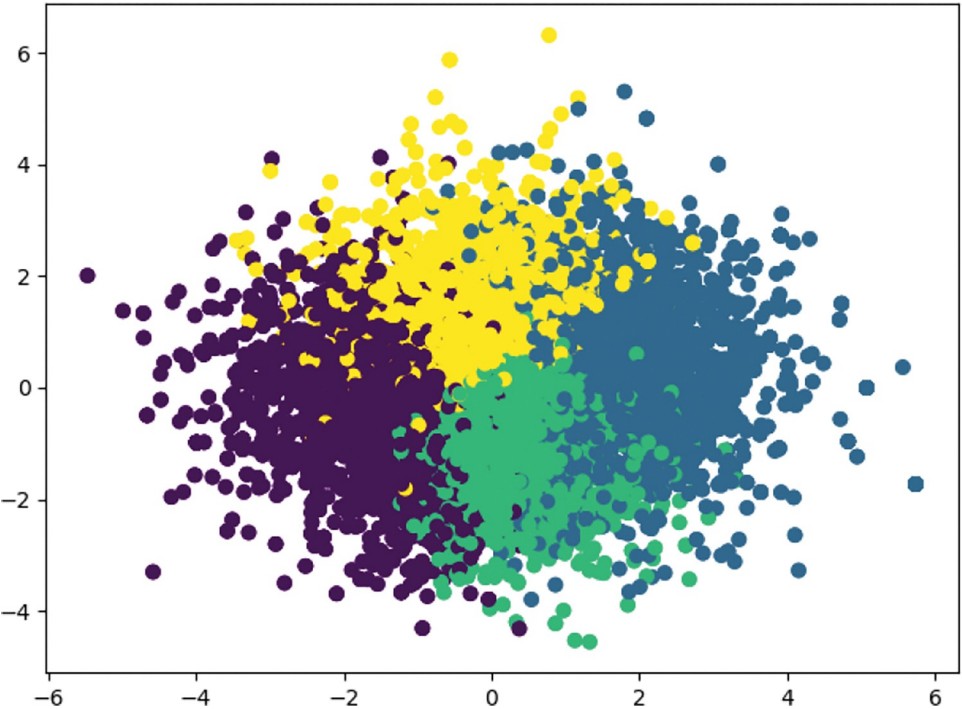

**Fig 4. Combined results of K-means clustering for both Jingdong and Taobao.**

Four clusters of influencing factors of OCPB can be clearly identified in the analyses of the Jingdong dataset, Taobao dataset, and combined Jingdong and Taobao dataset. After checking the context of four clusters, even though small differences were found, their influence was found to be negligible for our analyses. Thus, Fig 4 was chosen as the benchmark of influencing factors of OCPB. In Section 4.3, the explanation and analysis of influencing factors of OCPB will be presented.

## 4.2 Performance metrics

First, performance metrics of sum of the square errors (SSE) and silhouette coefficient were adapted to verify the clustering results of K-means.

When the number of clusters does not reach the optimal numbers K, SSE decreases rapidly with the increase of the number of clusters, while SSE decreases slowly after reaching the optimal numbers, and the maximum slope is the optimal numbers K.

$$SSE = \sum_{i=1}^{k} \sum_{p \in C_i} |p - m_i|^2 \tag{3}$$

Where $C_i$ is the $i$th cluster, $p$ is the sample point in $C_i$ (the mean value of all samples in $C_i$), and SSE is the clustering error of all samples, which represents the quality of clustering effect.

Fig 5 indicates that the SSE decreases rapidly when K equals the number of four.

Meanwhile, the silhouette coefficient can be shown as:

$$sc_i = \frac{b_i - a_i}{\max(b_i, a_i)} \tag{4}$$

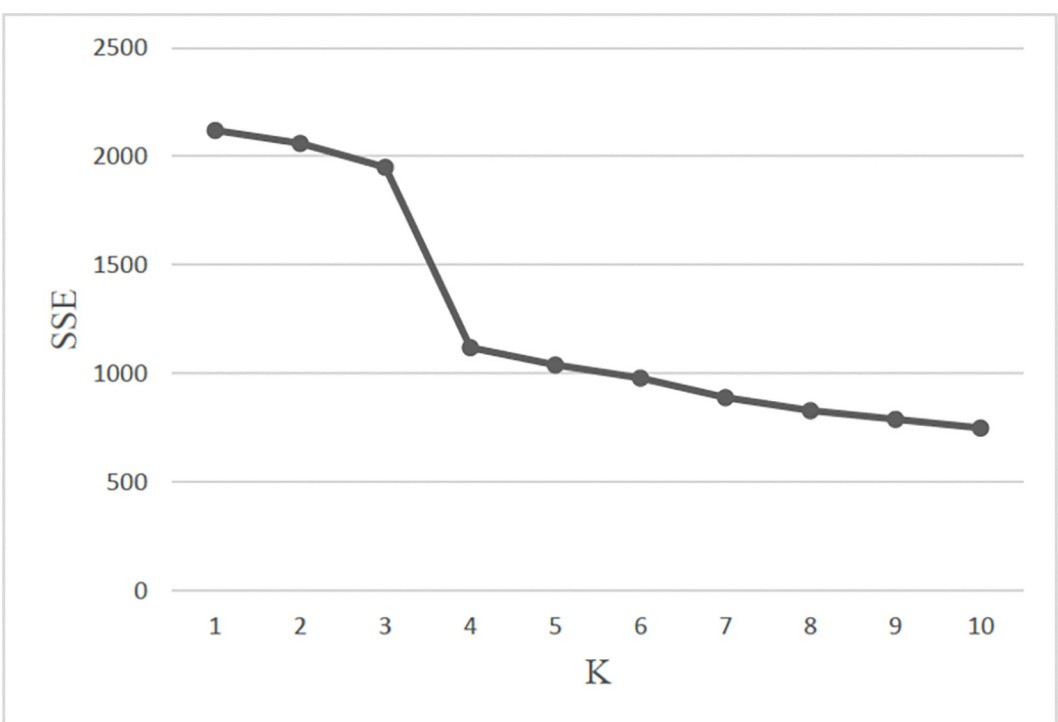

**Fig 5. SSE of K-means.**

The range of $sc_i$ is between -1 and 1, the clustering effect is bad when $sc_i$ is below zero, whereas the clustering effect is good when $sc_i$ is near 1 conversely.

Based on Fig 6, it is obviously to show that the silhouette coefficient reaches highest when K equals the number of four. Therefore, the results of the SSE and the silhouette coefficient jointly prove the number of K is four.

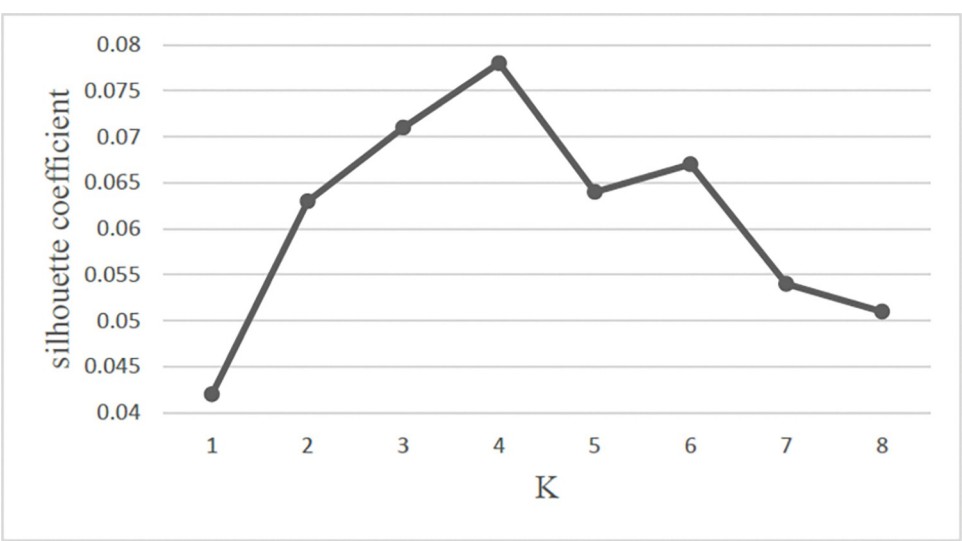

**Fig 6. Silhouette coefficient of K-means.**

### 4.3 Hypotheses results

Based on the K-means analysis, this section presents the influencing factors identified in the data from Jingdong and Taobao, which indicate the influencing factors influencing OCPB.

The first cluster comprises the perceived emergency context attribute, such as logistics, expressage, delivery, customer service, promotion, and reputation.

The second cluster comprises the perceived product attribute, such as appearance, brand, hand feeling, color, cost-performance ratio, price, design, and usability.

The third cluster comprises the perceived innovation attribute, such as photograph, quality and effects, screen quality, audio and video quality, pixel density, image resolution, earphone capabilities, and camera specifications.

The fourth cluster comprises the influencing factors, such as processing speed, operation, standby time, battery, system, internal storage, chip, performance, and fingerprint and face recognition, which cannot represent the perceived motivation attribute.

The results match the findings of Zhang et al. [70] to some extent, who identified 11 smartphone attributes based on online reviews: performance, appearance, battery, system, screen, user experience, photograph, price, quality, audio and video, and after-sale service. In addition, other scholars have explained the relationship between feature preferences and customer satisfaction [71, 72], usage behavior and purchase [73, 74], importance and costs of smartphones' features and services [75], brand effects [76], and purchase behavior of people of different ages and gender groups [77–79]. Thus, H1, H2 and H3 are supported, while H4 is not supported according to the results of the K-means analysis.

### 4.4 Theoretical framework and validity of OCPB influencing factors

Kotler's five product level model states that consumers have five levels of need comprising the core level, generic level, expected level, augmented level, and potential level. First, the core benefit is the fundamental need or want that consumers satisfy by consuming a product or service. Second, the generic level is a basic version of a product made up of only those features necessary for it to function. Third, the expected level includes additional features that the consumer might expect. Fourth, the augmented level refers to any product variations or extra features that might help differentiate a product from its competitors and make the brand a preferred choice amongst its competitors. Finally, a potential product includes all augmentations and improvements that a product might experience in the future [80].

In contrast with these levels, this study proposed the four influencing factors of OCPB. Based on Table 3, first, the perceived emergency context H1 is not included in Kotler's five products level, while the influencing factor expresses the significant characteristics of OCPB compared with Kotler's model. Second, the perceived product attribute H2 could be considered the core and generic level. Third, the perceived innovation attribute H3 could be considered the potential level. Fourth, the results of H4 mainly reflects additional or special function of product, which meets the definition of the expected and augmented level. To refine the theoretical framework, H4 changes to the perceived functionality attribute by combing the explanation of the expected and augmented level, instead of the perceived motivation attribute. The details are shown in Fig 7.

Fig 7 shows the four influencing factors of the theoretical framework of OCPB. Specifically, according to Kotler's five products level, the perceived product attribute is the necessary influencing factor of OCPB, which meets the core drive and basic requirement. For instance, the core drive of purchasing of a smartphone is the core function of communication, and then the appearance, brand, color, etc. The perceived functionality attribute is the additional influencing factor of OCPB, which meets the expected and augmented requirement. For instance,

**Table 3. OCPB hypotheses results.**

| Influencing factors | Hypotheses results | Examples (Statistical Frequency*) |
|---|---|---|
| Perceived emergency context attribute | H1 support | E-commerce (14597), take delivery of goods (12104), logistics (8611), expressage (8552), customer service (7256), packaging (5940), promotion (1246), and reputation (662) |
| Perceived product attribute | H2 support | Brand (11163), appearance (10045), hand feel (7825), satisfaction (3335), color (2513), cost-performance ratio (1638), price (1453), design (817), and usability (535) |
| Perceived innovation attribute | H3 support | Photograph (13129), quality and effects (11766), screen quality (11597), audio and video quality (7347), camera (1883), pixel density (1386), picture (890), image resolution (778), earphone capabilities (707), and model (641) |
| Perceived motivation attribute | H4 Not Support | Speed (12241), operation (11573), standby time (7116), battery (2773), system (2703), internal storage (1178), chip (1022), performance (982), fingerprint and face recognition (904), and quick charge (636) |

*Statistical Frequency: Only numbers greater than 500 are counted

when smartphones are in the same price range, consumers prefer to choose a smartphone belonging to better quality, smarter design, or better functionality. Moreover, the perceived innovation attribute is the attractive influencing factor of OCPB, which reflects the potential level. For instance, most consumers are the Apple fans mostly because the Apple products offer innovative usage experience and different technology elements yearly. Finally, the perceived emergency context attribute is the adaptive influencing factor of OCPB, which shows the main distinction with Kotler's five products level. Further, because of the COVID-19, consumers only have online channel to purchase product under a prolonged quarantine and lockdown. Thus, in the emergency context, consumers primarily consider whether the product can be purchased in the e-commerce platform, whether the product can be delivered normally, or whether the packaged has been disinfected fully.

## 5. Discussion

Traditional consumer behavior is mainly affected by psychological, social, cultural, economic, and personal factors [81, 82]. Park and Kim [83] conducted an empirical study to identify the key influencing factors that impact OCPB, which include service information quality, user interface quality, security perception, information satisfaction, and relational benefit. Further, Sata [84] conducted an empirical study and found that price, social group, product features, brand name, durability and after-sales services were important to consumers' buying behavior when choosing a smartphone for purchase. Simultaneously, some studies have utilized big data technology to explore OCPB, exploring online consumers' attitude toward products in different countries, and identified product features. However, these studies do not identify the influencing factors of OCPB and ignore e-WOM. To better explain OCPB influencing factors, e-WOM should be integrated into the theoretical framework and used in practical applications. Thus, this study contributes to OCPB research by data mining and analysis that can adequately identify the influencing factors based on e-WOM.

### 5.1 Theoretical implications

First, perceived emergency context attribute is the influencing factor of OCPB. Because of the COVID-19, e-commerce is the priority choice for consumers under circumstances of

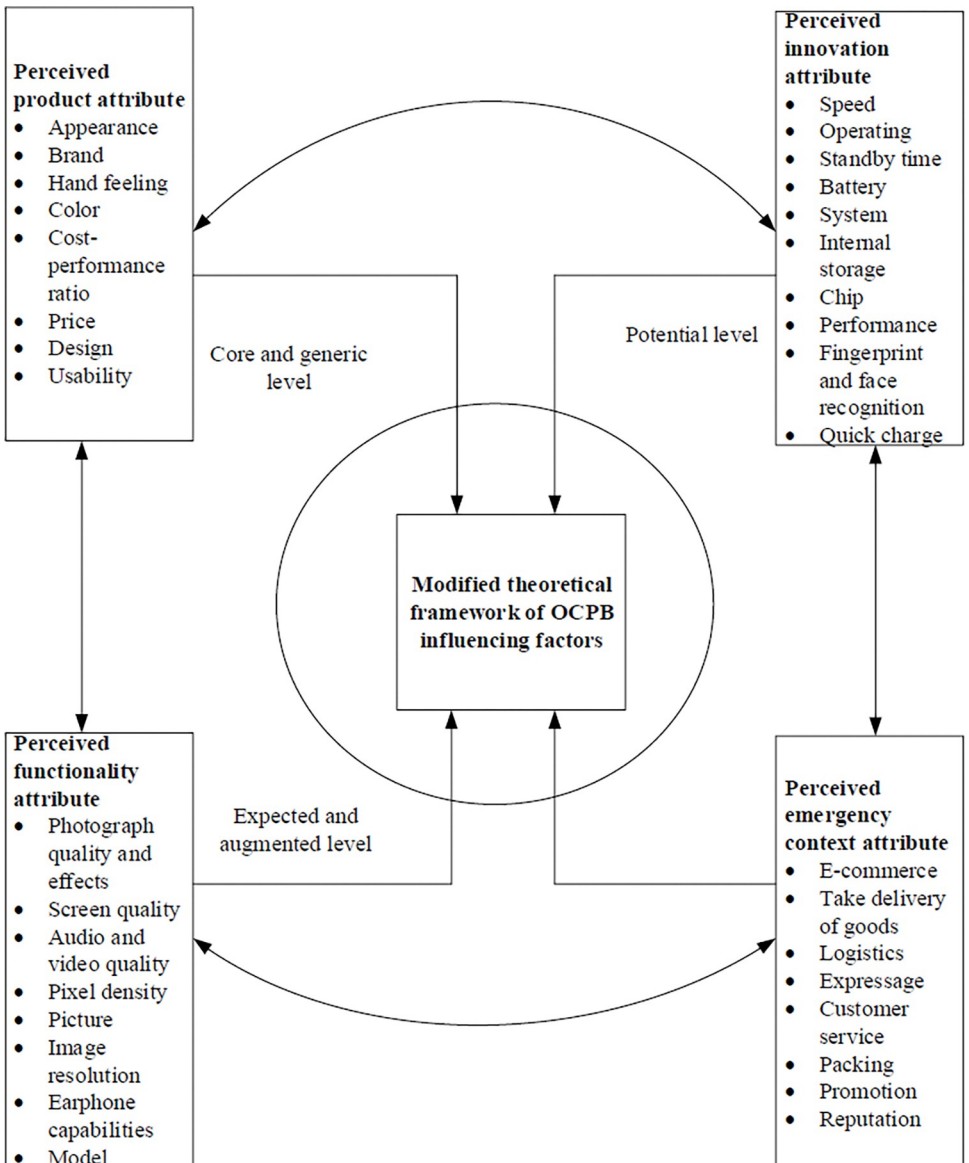

**Fig 7. The theoretical framework of OCPB influencing factors.**

prolonged quarantine and lockdown, and then considering logistics and delivery. Furthermore, customer service, packaging, promotion, and reputation are critical to online consumers.

Second, perceived product attribute is the influencing factors of OCPB. The basic features of product, such as appearance, brand, hand feeling, price, and design, positively attract online consumers. Elegant appearance, famous brand, better hand feeling, lower price, and better design would be more impactful to OCPB.

Third, perceived innovation attribute is the influencing factor of OCPB. For smartphone, online consumers would show more interest in the innovation of speed, operation, standby time, chip, etc. Scientific and technological innovation for most products could improve the level of OCPB. Thus, the guarantee and improvement of functionality of a product could create more opportunities for online consumers to make purchasing decisions.

Fourth, according to Kotler's five products level, perceived product attribute satisfies the characteristics of core drive and basic, while the perceived innovation attribute satisfies the characteristics of the potential level. Because hypothesis of perceived motivation attribute is not supported. Based on the analyzing results, the perceived functionality attribute is refined instead of the perceived motivation functionality attribute, which satisfies the expected and augmented. Meanwhile, the perceived emergency context attribute is not included, which shows the main difference with Kotler's five products level.

## 5.2 Managerial implications

The influencing factors of OCPB were clustered into four categories: perceived emergency context, product, innovation, and function attributes. The definition and explanation of these categories may have important managerial implications for both OCPB and e-commerce. First, the findings of this study suggest that e-commerce enterprises should pay more attention to improving the quality, user experience, and additional design features of their products to arouse the interest of OCPB. However, this may be difficult for e-commerce enterprises because achieving these goals requires updating the software and hardware constantly, which involves significant investment. For most scientific and technical corporations, making heavy investments is not particularly difficult, however, service-type enterprises and small and medium enterprises may have insufficient funds to afford such heavy investments. This is the main reason that most online consumers buy products from famous brands instead of small and medium enterprises. Therefore, to improve their situation, both types of companies could jointly develop products or services, for instance, small and medium enterprises may purchase patents from large enterprises, jointly researching and developing products, or large enterprises could share their achievements at a price.

Second, the pandemic has accelerated the spread of e-commerce considerably, changing consumers' shopping style in the process. Accordingly, e-commerce enterprises should adapt their marketing strategies, especially as the COVID-19 pandemic is still ongoing, due to the rapid development of the economy and its dynamic environment. For instance, e-commerce platforms should realize that changes in OCPB will continue to contribute to the growth of the e-commerce market. Moreover, e-commerce enterprises should combine their online presence with brick-and-mortar stores. Even more importantly, e-commerce enterprises should successfully operate their supply chain to adapt to the implementation of lockdown measures and the closing of manufacturing factories. Consumers should exercise caution when facing e-commerce enterprises' adaptive financial policy, such as interest-free rates, which may cause financial burden.

Third, e-commerce enterprises should offer a simple and smooth shopping experience, clearly display practical information, increase the value of goods (by improving the quality, design, and performance of products or services) and improve their brand image for online consumers. However, e-commerce enterprises sometimes rely on certain fraudulent methods to increase their sales volume, such as falsifying positive e-WOM and deleting negative feedback, as was identified during the data processing stage. Therefore, online consumers should select online stores cautiously to avoid buying products of poor quality or performance.

Fourth, nowadays, technology is constantly evolving at an accelerated rate, particularly in the smartphone industry, as companies launch new products with innovative functions each year. Thus, e-commerce enterprises should strive to innovate to secure their position in the market. In addition, consumers should reconsider the need to experience the state-of-the-art products because these may have high prices.

## 6. Conclusion and limitations

In conclusion, during the COVID-19 pandemic, consumers highly preferred to buy products online, because most brick-and-mortar stores were closed due to lockdowns and social distancing measures. Additionally, with the rapid development of e-commerce, online shopping has become the most popular shopping style because it allows consumers to not only save time and money, but also review e-WOM before purchasing a product. Moreover, e-WOM is much more reliable compared with traditional WOM. Thus, this study proposed a theoretical framework to explore and define the influencing factors of OCPB based on e-WOM data mining and analyzing. The data were crawled from Jingdong and Taobao, while the data process was also fully demonstrated. Comparing the results, the influencing factors of OCPB were clustered around four categories: perceived emergency context, product, innovation, and function attributes. Moreover, perceived emergency context attribute is the main difference compared with Kotler's five products level, while perceived product attribute meets the core and generic level, perceived functionality attribute meets the expected and augmented level, and perceived innovation attribute meets the potential level.

However, this study still has certain limitations. First, the data were crawled from Chinese e-commerce websites, hence, they may not be generalized in contexts where the influencing factors and dimensions may vary compared with other countries or regions. Second, this study only explored and defined the antecedents of OCPB. Data should be added from Western e-commerce websites. Moreover, the present study's results should be compared with Western studies to generate a more comprehensive view of the antecedents of OCPB. Future studies should explore the underlying mechanisms influencing OCPB.

## Supporting information

**S1 Dataset.**
(XLSX)

## Author Contributions

**Conceptualization:** Xiaoya Zhu.

**Data curation:** Manman Wang.

**Formal analysis:** Fuli Zhou, Shuang Cheng.

**Funding acquisition:** Qiwei Wang, Xiaoya Zhu, Manman Wang, Fuli Zhou, Shuang Cheng.

**Investigation:** Qiwei Wang.

**Methodology:** Qiwei Wang.

**Project administration:** Qiwei Wang, Xiaoya Zhu, Manman Wang, Fuli Zhou, Shuang Cheng.

**Resources:** Qiwei Wang.

**Software:** Qiwei Wang.

**Supervision:** Qiwei Wang, Xiaoya Zhu.

**Validation:** Qiwei Wang.

**Visualization:** Qiwei Wang.

**Writing – original draft:** Qiwei Wang.

**Writing – review & editing:** Qiwei Wang.

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
