## [Decision Letter · Decision Letter 0]

2 May 2023

PONE-D-23-11851A Theoretical Model of Factors Influencing Online Consumer Purchasing Behavior through Electronic Word of Mouth Data Mining and AnalysisPLOS ONE

Dear Dr. ZHU,

Thank you for submitting your manuscript to PLOS ONE. After careful consideration, we feel that it has merit but does not fully meet PLOS ONE’s publication criteria as it currently stands. Therefore, we invite you to submit a revised version of the manuscript that addresses the points raised during the review process.

We look forward to receiving your revised manuscript.

Kind regards,

Ahmad Samed Al-Adwan

Academic Editor

PLOS ONE

Journal Requirements:

   "The author(s) disclosed receipt of the following financial support for the research, authorship, and/or publication of this article: This study was supported by the Henan Province Philosophy and Social Science Planning Project (grant number. 2020CZH012), the Henan Key Research and Development and Promotion Special (Soft Science Research) (grant number. 222400410126), the Jiangsu Province Social Science Foundation Youth Project (grant number. 21GLC012) and the Doctor Fund of Zhengzhou University of Light Industry (grant number. 2020BSJJ022, 2019BSJJ017).Conceptualization: Xiaoya Zhu.

Data curation: Manman Wang.

Formal analysis: Fuli Zhou, Shuang Cheng.

Funding acquisition: Qiwei Wang, Xiaoya Zhu, Manman Wang, Fuli Zhou, Shuang Cheng

Investigation: Qiwei Wang.

Methodology: Qiwei Wang.

Project administration: Xiaoya Zhu, Manman Wang, Fuli Zhou, and Shuang Cheng

Resources: Qiwei Wang.

Software: Qiwei Wang.

Supervision: Qiwei Wang, Xiaoya Zhu, Manman Wang, Fuli Zhou, Shuang Cheng

Writing – original draft: Qiwei Wang.

Writing – review & editing: Qiwei Wang."

5. Please ensure that you include a title page within your main document. You should list all authors and all affiliations as per our author instructions and clearly indicate the corresponding author.

Additional Editor Comments:

I am pleased to inform you that we have reviewed your manuscript and find it to be of high quality and relevance to our journal's scope. Congratulations on your excellent work! However, to ensure that your manuscript meets the highest standards of academic excellence, we recommend that you carefully address the reviewers' comments, which we have attached with this email. The comments highlight some minor revisions that can enhance the clarity and impact of your work. We kindly request that you carefully consider each of the reviewers' comments and address them appropriately in your revised manuscript. Once you have made the necessary changes, please resubmit your manuscript through our online submission system.

Reviewers' comments:

Reviewer's Responses to Questions

**Comments to the Author**

1. Is the manuscript technically sound, and do the data support the conclusions?

Reviewer #1: Yes

Reviewer #2: Yes

2. Has the statistical analysis been performed appropriately and rigorously? 

Reviewer #1: Yes

Reviewer #2: Yes

3. Have the authors made all data underlying the findings in their manuscript fully available?

Reviewer #1: Yes

Reviewer #2: Yes

4. Is the manuscript presented in an intelligible fashion and written in standard English?

Reviewer #1: Yes

Reviewer #2: Yes

5. Review Comments to the Author

Reviewer #1: As a journal reviewer, I would like to commend the authors for their rigorous data analysis process and the comprehensive results presented in the article. However, there are two minor aspects that need to be addressed by the authors in order to increase the quality of this paper.

The introduction section can be strengthening by clearly highlighting the research problem, significance, and gaps in the literature. This would help outline the main contributions of this submission.

It has been noticed that you have cited a bit old research (2010, 2011, 2013). You are suggested to update these citations with recent and well-established ones.

Reviewer #2: This is a well-written paper with good contributions. All important elements of a good academic research paper are presented in this submission. I would suggest a few minor amendments.

1. The abstract appears to be more focused on reporting results rather than effectively communicating the central idea(s) explored in the paper and the contribution it makes to the scientific community's existing knowledge. Its lack of coherence makes it challenging to follow.

2. The discussion section should first compare the findings of this research with those of prior research. Both theoretical and practical implications should be reported in a separate section.

3. The literature review, particularly the literature related to the SOR and WOM, should be strengthen by including well-established and related research.

This includes but is not limited to:

- E-commerce in high uncertainty avoidance cultures: The driving forces of repurchase and word-of-mouth intentions. Doi: https://doi.org/10.1016/j.techsoc.2022.102083

- Boosting Online Purchase Intention in High-Uncertainty-Avoidance Societies: A Signaling Theory Approach. Doi: https://doi.org/10.3390/joitmc8030136

- Solving the product uncertainty hurdle in social commerce: The mediating role of seller uncertainty. Doi: https://doi.org/10.1016/j.jjimei.2023.100169

- Boosting Customer E-Loyalty: An Extended Scale of Online Service Quality. Doi:https://doi.org/10.3390/info10120380

- Building customer loyalty in online shopping: the role of online trust, online satisfaction and electronic word of mouth. Doi: https://doi.org/10.1504/IJEMR.2020.108132

6. PLOS authors have the option to publish the peer review history of their article (what does this mean?). If published, this will include your full peer review and any attached files.

Reviewer #1: **Yes: **Malek alsoud

Reviewer #2: **Yes: **Dr Husam Yaseen

---

## [Author Response · Author response to Decision Letter 0]

5 May 2023

Dear Editors and Reviews:

 Thank you for your correspondence and for the feedback provided by the reviewers regarding our manuscript titled “A Theoretical Model of Factors Influencing Online Consumer Purchasing Behavior through Electronic Word of Mouth Data Mining and Analysis”. These valuable and constructive comments have assisted in revising and enhancing our paper, in addition to providing significant guidance to our research. We have carefully examined and addressed all provided comments, and hope that the corrections meet with approval. The main revisions made to the paper and our corresponding responses to the reviewers' comments are listed below.

 Responds to the Journal Requirements:

Response: We have modified our manuscript to ensure compliance with PLOS ONE's style requirements, including those for file naming.

Response: The cover letter contains a statement indicating that "The funders had no role in study design, data collection and analysis, decision to publish, or preparation of the manuscript."

3. In your Data Availability statement, you have not specified where the minimal data set underlying the results described in your manuscript can be found. PLOS defines a study's minimal data set as the underlying data used to reach the conclusions drawn in the manuscript and any additional data required to replicate the reported study findings in their entirety. All PLOS journals require that the minimal data set be made fully available.

Response: We have uploaded a separate file consisting of the study's minimal data.

4. PLOS requires an ORCID iD for the corresponding author in Editorial Manager on papers submitted after December 6th, 2016. Please ensure that you have an ORCID iD and that it is validated in Editorial Manager.

Response: The corresponding author has updated the ORCID iD.

5. Please ensure that you include a title page within your main document. You should list all authors and all affiliations as per our author instructions and clearly indicate the corresponding author.

Response: we have included a title page with information for all authors in our main document.

6. Please review your reference list to ensure that it is complete and correct. If you have cited papers that have been retracted, please include the rationale for doing so in the manuscript text, or remove these references and replace them with relevant current references.

Response: Our reference list has been carefully reviewed and verified to be both complete and accurate.

 Special thanks to you for your good comments.

 Responds to the reviewers’ comments:

Reviewer #1: As a journal reviewer, I would like to commend the authors for their rigorous data analysis process and the comprehensive results presented in the article. However, there are two minor aspects that need to be addressed by the authors in order to increase the quality of this paper.

The introduction section can be strengthening by clearly highlighting the research problem, significance, and gaps in the literature. This would help outline the main contributions of this submission.

It has been noticed that you have cited a bit old research (2010, 2011, 2013). You are suggested to update these citations with recent and well-established ones.

Response: Firstly, the introduction section has been rewritten to address the reviewers' comments. The research problem, significance, and gaps in the literature have all been included in the introduction. Secondly, outdated references from 2010, 2011, and 2013 have been removed and replaced with more recent references.

Special thanks to you for your good comments.

Reviewer #2: This is a well-written paper with good contributions. All important elements of a good academic research paper are presented in this submission. I would suggest a few minor amendments.

1. The abstract appears to be more focused on reporting results rather than effectively communicating the central idea(s) explored in the paper and the contribution it makes to the scientific community's existing knowledge. Its lack of coherence makes it challenging to follow.

 Response: The abstract section has been rewritten to enhance the logical flow and coherence of the paper, as suggested by the reviewers.

2. The discussion section should first compare the findings of this research with those of prior research. Both theoretical and practical implications should be reported in a separate section.

Response: In the discussion section, we have compared the findings of our research with those of prior research, and have separated the theoretical and practical implications into distinct sections.

3. The literature review, particularly the literature related to the SOR and WOM, should be strengthen by including well-established and related research.

This includes but is not limited to:

- E-commerce in high uncertainty avoidance cultures: The driving forces of repurchase and word-of-mouth intentions. Doi: https://doi.org/10.1016/j.techsoc.2022.102083

- Boosting Online Purchase Intention in High-Uncertainty-Avoidance Societies: A Signaling Theory Approach. Doi: https://doi.org/10.3390/joitmc8030136

- Solving the product uncertainty hurdle in social commerce: The mediating role of seller uncertainty. Doi: https://doi.org/10.1016/j.jjimei.2023.100169

- Boosting Customer E-Loyalty: An Extended Scale of Online Service Quality. Doi:https://doi.org/10.3390/info10120380

- Building customer loyalty in online shopping: the role of online trust, online satisfaction and electronic word of mouth. Doi: https://doi.org/10.1504/IJEMR.2020.108132

Response: Additionally, we have incorporated the listed references suggested by the reviewer to strengthen the literature review.

Special thanks to you for your good comments.

 We have made our best effort to improve the manuscript and greatly appreciate the Editors and Reviewers' earnest work. We hope that the revised manuscript meets with approval.

Thank you once again for your valuable comments and suggestions.

 Best regards,

Dr. Xiaoya Zhu

---

## [Editor Report · Decision Letter 1]

7 May 2023

A Theoretical Model of Factors Influencing Online Consumer Purchasing Behavior through Electronic Word of Mouth Data Mining and Analysis

PONE-D-23-11851R1

Dear Dr. ZHU,

We’re pleased to inform you that your manuscript has been judged scientifically suitable for publication and will be formally accepted for publication once it meets all outstanding technical requirements.

Kind regards,

Ahmad Samed Al-Adwan

Academic Editor

PLOS ONE
---

## [Editor Report · Acceptance letter]

10 May 2023

PONE-D-23-11851R1 

A Theoretical Model of Factors Influencing Online Consumer Purchasing Behavior through Electronic Word of Mouth Data Mining and Analysis 

Dear Dr. ZHU:

I'm pleased to inform you that your manuscript has been deemed suitable for publication in PLOS ONE. Congratulations! Your manuscript is now with our production department. 

Kind regards, 

on behalf of

Prof. Ahmad Samed Al-Adwan 

Academic Editor

PLOS ONE